# Polygenic Risk Score Implementation into Clinical Practice for Primary Prevention of Cardiometabolic Disease

**DOI:** 10.3390/genes15121581

**Published:** 2024-12-09

**Authors:** Julia Hughes, Mikayla Shymka, Trevor Ng, Jobanjit S. Phulka, Sina Safabakhsh, Zachary Laksman

**Affiliations:** 1Department of Medicine, University of British Columbia, Vancouver, BC V5M 1M9, Canada; juliahug@student.ubc.ca (J.H.); mikayls5@student.ubc.ca (M.S.); trevorng@student.ubc.ca (T.N.);; 2Centre for Heart Lung Innovation (HLI), Vancouver, BC V6Z 1Y6, Canada; 3School of Biomedical Engineering, University of British Columbia, Vancouver, BC V6T 2B9, Canada

**Keywords:** polygenic risk scores, primary prevention, cardiometabolic

## Abstract

**Background**: Cardiovascular disease is a leading cause of mortality globally and a major contributor to disability. Traditional risk factors, as initially established in the FRAMINGHAM study, have helped to stratify populations and identify patients for early intervention. Incorporating genetic factors enhances risk stratification tools, enabling the earlier identification of individuals at increased risk and facilitating more targeted and effective risk factor modifications. While monogenic risk variants are present in a minority of the population, polygenic risk scores (PRS) are collections of multiple single-nucleotide variants that collectively provide summative risk and capture a more accurate risk score for a greater number of people. PRS have demonstrated clear utility in cardiometabolic diseases by predicting onset, progression, and therapeutic response. **Methods**: A structured and exploratory hybrid search strategy was employed, combining keyword-based database searches and supplementary techniques to comprehensively synthesize the literature on PRS implementation in clinical practice. **Discussion**: A comprehensive overview of PRS in cardiometabolic diseases and their potential avenues for integration into primary care is discussed. First, we examine the implementation of genetic screening, risk communication, and intervention strategies through the lens of the American Heart Association’s implementation criteria, focusing on their efficacy, minimization of harm, and logistical considerations. Then, we explores how the varied perceptions of patients and practitioners towards PRS can influence both adoption and utilization. Lastly, we addresses the need for the development of clear guidelines and regulations to support this process, ensuring PRS integration is both scientifically sound and ethically responsible. **Future directions**: Initiatives aimed at advancing personalized approaches to disease prevention will enhance health outcomes. Developing guidelines for the responsible use of PRS by establishing benefits, while mitigating risk, will a key factor in implementation for clinical utility. **Conclusions**: For integration into clinical practice, we must address both patient and provider concerns and experience. Standardized guidelines and training will help to effectively implement PRS into clinical practice. Developing these resources will be essential for PRS to fulfill its potential in personalized, patient-centered care.

## 1. Introduction

### 1.1. Risk Factors and Traditional Use

The foundation of cardiovascular disease (CVD) risk assessment in clinical practice stems from the landmark Framingham Heart Study, initiated in 1948 to investigate the epidemiology of heart disease. In a pivotal early publication, Kannel coined the term “risk factor” to describe lifestyle, environmental, and biological factors that appeared to precede the development of heart disease [1]. This concept became instrumental in identifying factors that contribute to CVD, transforming how risk is perceived and managed.

The Framingham study findings led to the development of the Framingham Risk Score (FRS), a tool that quantifies an individual’s CVD risk based on several established factors including age, sex, diabetes status, smoking, total cholesterol, HDL cholesterol, and blood pressure [2]. The FRS has become the gold standard in risk stratification, shaping CVD risk assessments by emphasizing modifiable risk factors and promoting targeted interventions such as blood pressure control, cholesterol management, and smoking cessation.

However, traditional risk models focus heavily on age-related and lifestyle factors, overlooking genetic predisposition that contributes to lifelong CVD risk. These models often focus on risk factors that emerge or worsen over time [3], while genetic susceptibility provides a more nuanced understanding by considering lifelong genetic influence [4]. However, it is important to note that some factors incorporated in this risk score do have widely established genetic underpinnings, such as diabetes and HDL cholesterol. Integrating genetic risk with acquired risk could enhance risk prediction, maintaining a focus on primary prevention while highlighting genetic vulnerability [5,6]. 

### 1.2. What Is a PRS?

Polygenic risk scores (PRS) leverage data from genome-wide association studies (GWAS) to assess disease risk based on multiple genetic variants within an individual’s genome. These scores are computed by integrating single-nucleotide variants (SNVs) with their effect sizes [7,8,9,10]. In polygenic disorders, individual variants lack sufficient predictive power alone. However, researchers have demonstrated that combining multiple risk variants into a PRS has promising results in disease prediction. Each SNV’s contribution is weighted by its effect size, resulting in a single numerical indicator of genetic susceptibility for a disease or trait [8]. Importantly, not all SNVs increase disease risk; some are protective, decreasing susceptibility to certain conditions, and their effects must also be integrated into the PRS for a comprehensive assessment [11]. There is also the consideration that not all SNVs may have a biological impact. Uncertainty exists regarding whether some SNVs have an important biological role, even with a high statistical significance, given the unknown effect sizes [11].

Some research even suggests PRS can help explain the variable expressivity of certain well-known monogenic familial diseases [8]. Since any given SNV may exhibit correlation with others, methods have been adopted to account for these correlations to produce a more accurate PRS. These methods are varied and can be as simple as removing a random SNV from a pair that is clearly correlated. More complex PRS calculations can retain all SNVs but reweigh them according to their effect sizes [7,9]. Furthermore, the accuracy and applicability of PRS are heavily influenced by the ancestry of the individual, as allele frequencies and effect sizes of SNVs differ significantly across populations [12]. PRS developed for one ancestral group may perform poorly in another, underscoring the need for ancestry-informed methods and diverse GWAS datasets to ensure equitable and precise predictions [13].

PRS methodology assumes an additive genetic model, where risk and protective alleles across variants are summed to quantify genetic predisposition. Despite concerns about validity, PRS have demonstrated their ability to predict disease status effectively in various study designs, including case–control studies and population-based cohorts. These scores are particularly useful in polygenic disorders where no single variant provides sufficient risk assessment alone. They enhance predictive accuracy when combined with traditional clinical risk factors, such as age and lifestyle habits [8].

Ongoing research explores advanced PRS calculation methods, including Bayesian approaches that model variant correlations more explicitly [7]. Such developments aim to enhance the predictive power of PRS, refine their application in identifying disease susceptibility, and ensure their utility across diverse ancestral groups [10].

### 1.3. Current Clinical Utility

The interpretation of PRS varies by disease but generally suggests that higher risk scores may predict disease onset, progression, and treatment response [7]. Therefore, it is proposed that PRS use for screening facilitates identification of individuals at an earlier stage of disease, allowing for earlier intervention and improved outcomes. Early identification of at-risk individuals using PRS may motivate lifestyle changes and decrease the need for pharmacologic intervention [8,14,15,16].

PRS-based risk prediction models are often being used in conjunction with existing clinical risk factor models to improve prediction power and refine indications for treatment [7,17,18]. While many of these models are in early stages and being used to demonstrate the potential utility of PRS in clinical practice [7], few studies have investigated their real-world use, highlighting a need for further research to continue to advance PRS from development to practical applications.

The American Heart Association (AHA) proposes clinical utilities of PRS for a variety of cardiometabolic diseases, including early screening, both pharmacological and non-pharmacological interventions, and personalized drug therapies [19]. Within this review, we discuss the implementation of PRS broadly in cardiometabolic diseases, and refer to more specific descriptions of cardiometabolic disease, such as CVD, when applicable. Currently, there are longitudinal efforts being undertaken in order to further define the clinical utility of PRS for several common diseases [20,21,22,23]. Current challenges in clinical PRS trials include population structure adjustments, result interpretation, and feasibility of integration, including logistics, and evaluation of patient and provider experiences [20,21,22,23]. However, with genetic testing becoming more affordable and research on cardiometabolic PRS continuing to expand, it is likely that PRS will integrate with clinical practice [24], emphasizing the need to address these challenges. 

In its scientific statement on PRS, the AHA outlines the following three criteria that need to be broadly considered before PRS can be implemented into healthcare systems: (1) Efficacy, (2) Harm, and (3) Logistics [19]. In this review, we will consider these three criteria as we explore PRS implementation into clinical practice and discuss implementation in screening genetics, patient and physician attitudes, and a discussion of guidelines for PRS integration. Figure 1 illustrates the relevant shareholders that are included in our review.

## 2. Materials and Methods

To provide a comprehensive overview of the literature related to PRS implementation in clinical practice, a structured but flexible search strategy was utilized. Initially, PubMed and Medline were used to conduct a search using a combination of key terms related to polygenic risk score implementation, e.g., “Polygenic risk score”, “Monogenic risk”, “Clinical risk”, “Patient Perspectives”, “Provider Attitudes”, “Clinical guidelines”, “Ethics”, “Implementation”, and “Integration” and related terms to allow for relevant studies to be more broadly captured. Further, an exploratory approach was employed to more completely understand the topic; this included consulting major journals and search engines for supplementary articles. Reference list searching was also conducted in relevant articles in order to identify any additional sources not captured in the initial search. This hybrid approach allowed a focused synthesis of the current literature.

## 3. Discussion

### 3.1. PRS in Screening Genetics

PRS are deemed suitable when they significantly enhance the accuracy of clinical risk assessment tools or when they identify individuals at comparable risk levels to those with monogenic risk variants [7]. Integration of PRS into currently established risk models for several cardiometabolic conditions including atrial fibrillation (AF), coronary artery disease (CAD), and type 2 diabetes (T2D) consistently improves predictive accuracy beyond established clinical factors [7,25,26,27].

In clinical settings, PRS implementation has shown promise in cohorts with higher disease probabilities, aiding early diagnosis or informing treatment decisions [28].

One study demonstrated that adding PRS to conventional risk models enhances risk discrimination for CVD. This addition increases the Concordance Index, a measure of how accurately patients are sorted based on event occurrence, modestly when compared to a conventional prediction model, leading to substantial improvements in risk reclassification, benefitting both cases and non-cases of CVD [29].

While PRS offer promise in refining risk stratification and preventive strategies for common diseases, including cardiometabolic disease, their full clinical utility remains under evaluation [25,30]. CVDs are among the most commonly studied PRS at the moment and there are several trials evaluating the clinical efficacy of such scores [13,25,31]. The INNOPREV trial is the largest of such trials, enrolling over 1000 individuals with high CVD risk as defined through traditional assessments. This randomized control trial evaluates CVD risk communication through one of four approaches, including traditional methods or a combination of traditional methods and PRS [31]. Changes in lifestyle patterns and CVD risk profiles will be monitored over one year using scoring metrics, blood sample analyses, and repeated assessments. The trial explored PRS efficacy, with potential public health implications for reducing the burden of CVD on healthcare systems.

### 3.2. Mono- and Polygenic Risk Factors

Understanding the genetic factors of cardiometabolic disease involves examining the complex interactions between monogenic (Mendelian) and polygenic risk variants. These factors collectively contribute to disease susceptibility and progression, shaping contemporary approaches to risk assessment and personalized medicine [7].

Monogenic mutations associated with familial forms of diseases like familial hypercholesterolemia (FH) and early-onset autoimmune disorders represent well-established risk factors with clear inheritance patterns. Despite efforts to identify monogenic causes, a significant proportion of cases remain unexplained, prompting exploration into the role of polygenic influences [32]. Recent studies highlight the growing recognition that polygenic burden significantly modulates disease outcomes, particularly evident in conditions where traditional monogenic explanations are insufficient [8,32,33,34].

PRS, derived from GWAS, have emerged as valuable tools in cardiovascular risk prediction [13,31]. For example, high PRS values for CAD are associated with elevated LDL cholesterol levels and increased CAD risk, akin to being positive for FH but without presenting severe hypercholesterolemia. Integrating CAD PRS into clinical risk calculators enhances risk stratification across diverse populations, potentially identifying individuals overlooked by conventional risk assessment tools [7,8].

The utility of PRS extends beyond prediction to refining diagnostic strategies. Studies show that pre-screening individuals with a polygenic risk score for conditions like T2D can prioritize those likely to harbor monogenic variants, optimizing diagnostic sequencing and reducing unnecessary tests [35]. This approach underscores the complementary roles of polygenic and monogenic assessments in clinical practice, enhancing diagnostic accuracy and resource allocation [13,31].

Further insights into the genetics of CAD through large-scale studies have identified novel susceptibility loci in different populations and emphasized the importance of trans-ancestry meta-analyses in deriving more accurate PRS [13]. In one GWAS, eight new susceptibility loci and Japanese-specific rare variants were discovered to influence CAD severity and mortality. These new findings were incorporated into a PRS after a subsequent trans-ancestry meta-analysis, creating a novel PRS that outperformed previous PRS derived solely from Japanese or European GWAS [13]. Challenges persist in integrating PRS effectively with existing risk models such as the Pooled Cohort Equations for CAD. While the addition of PRS to these equations improves predictive accuracy modestly, questions remain regarding their widespread clinical implementation and validation across diverse populations [25]. 

The interplay between monogenic and polygenic risks raises fundamental questions about disease pathogenesis and treatment responses [30]. Evidence suggests that while monogenic variants may confer substantial disease risk, their penetrance and clinical impact can be influenced by underlying polygenic backgrounds [8,33]. Continued research into the interrelationships between monogenic and polygenic factors promises to refine our understanding of disease mechanisms and improve clinical strategies for CVD prevention and management. Notably, the identification of influential monogenic variants must be considered prior to broad application of PRS.

### 3.3. Model Accuracy

PRS have emerged as pivotal tools in stratifying CVD risk, particularly for conditions such as CAD, AF, and stroke [27,36]. Initially developed using data predominantly from European populations, PRS have demonstrated robust predictive accuracy for CAD in these populations, often outperforming individual clinical risk factors like smoking and hypertension [7]. Some studies suggest that PRS for CAD had stronger predictive accuracy in individuals where other traditional risk factors have yet to manifest [30].

This predictive power extends beyond mere risk stratification within clinical risk categories; PRS can effectively identify individuals at heightened risk who might benefit from targeted interventions such as statin therapy, thereby potentially enhancing preventive efforts and treatment outcomes [7,29]. In one prospective study, 7091 insulin-naive study participants were followed for progression onto insulin use. Traditional risk factors such as BMI, triglyceride levels and tobacco use as well as A1c targets were used in one cohort to establish associations with eventual insulin usage. A PRS using 123 known variants for T2DM was applied to a replication cohort and found to effectively predict rapid progression to insulin use. The PRS was also able to identify about 5% of study participants who failed to control their A1c while on two oral glycemic lowering drugs, an indication for early statin therapy with these individuals. These scores not only prioritize risk factors among clinical parameters but also segregate individuals with heightened long-term cardiovascular mortality risk, demonstrating their potential in guiding targeted preventive interventions [26].

Moreover, recent advancements in PRS development for conditions like aortic stenosis highlight ongoing efforts to enhance risk discrimination beyond traditional clinical factors. In one study, the genetic risk component in AS was found to predict disease in a fashion similar to traditional risk calculators. When combined with traditional factors, this PRS provided modest improvements in risk estimation. These efforts are crucial as they pave the way for integrating PRS into clinical practice, potentially improving patient outcomes through more precise risk prediction and targeted therapeutic strategies [18].

Nevertheless, challenges persist regarding the precision of PRS across different ancestry groups [7,8,30]. GWAS are predominantly those with European ancestry. This has given rise to the challenge of re-calibrating PRS for different ethnicities. Ongoing efforts to remedy these discrepancies are proving fruitful, with sparse amounts of newly adjusted PRS showing comparable predictive accuracy among different ethnicities [7,10,12,26]. While there are some studies concerned with African American and Chinese demographics and their subsets, the literature is notably lacking in PRS applications to many non-western European demographics. Efforts to expand genomic data collection and refine statistical methodologies in multi-ethnic cohorts are underway to address these disparities and improve the clinical utility of PRS universally [8].

Another issue of PRS accuracy lies in its lack of characterization amongst groups of the same genetic ancestry, but different environmental exposures [30]. Some of PRS’s predictive signals do not take confounding environmental factors into account. Epigenetics are currently not being incorporated into PRS. 

### 3.4. Perception of Patients and Providers

In order for PRS to be successfully implemented into clinical practice, it is essential to assess the perspectives of both patients and providers. 

Patient perspectives on the communication of PRS information is multifaceted. Studies have shown that disclosing genetic information to patients can increase their perception of personal control over their health, leading to greater self-efficacy and proactive health behaviors such as information seeking and sharing and lifestyle modifications [37,38,39,40]. These health behavior changes, such as seeking medical care, exercise, weight loss, and dietary changes, were particularly evident among individuals who were informed of a high-risk result [39,40,41,42]. Findings are mixed in terms of health outcomes as a result of behavior changes, with some support for improvement in health outcomes [40] and others finding no significant change in associated health outcomes despite behavior change [41].

Patients largely report that the main message conveyed by PRS is easily understandable to them [20,43,44], although for some, the numerical aspects of PRS representation were more difficult to understand [44]. Concern remains regarding the potential for negative emotional responses to result disclosure. The literature is divided as some studies indicate that result disclosure does not induce significant health anxiety and suggest that genetic risk communication is possible without psychological effects for patients [43]. However, it has also been found that patients with higher PRS had higher levels of negative feelings and uncertainty regarding PRS than patients that were informed they had lower risk scores [45]. Despite varying emotional responses, patients still generally believe that the benefits to their health outweigh the risks, supporting the implementation of PRS into clinical practice for the general public [20,45]. Notably, these findings are all in the context of cardiometabolic PRS. 

In terms of provider attitudes regarding PRS implementation, the majority of healthcare providers have found ease in incorporating the communication of genetic risk information to patients [20]. Providers reported that PRS are clinically useful, straightforward, and feasible to incorporate into routine care as demonstrated in studies where PRS were used to inform clinical management, motivate preventive behavior, and identify previously unknown high-risk patients [20,44]. Most providers support the integration of PRS into practice, especially for higher-risk patients [20,44,46,47].

Key concerns include the lack of standardized guidelines for PRS communication and ambiguity about actionable steps based on PRS results [48]. These concerns underscore the need for established clinical guidelines, further addressed in subsequent discussion. Other practical challenges that providers feel need to be addressed include concerns related to managing workflow and result responsibilities, resource requirements, and the need for training to effectively discuss genetic results with patients [44,47,48].

Overall, both patients and providers have largely favorable views when it comes to the integration of PRS into clinical practice, especially with its potential to motivate preventative health behaviors and guide management for high-risk patients. However, barriers must be addressed for consistent, effective PRS implementation, including the development of clear clinical guidelines, logistical workflow integration, and support for managing patient emotions and expectations. 

### 3.5. PRS Guidelines

Integration of PRS into clinical practice offers new potential for CVD risk prediction, but the development of standardized guidelines remains an outlying issue. Guidelines can facilitate PRS implementation into clinical utility by defining roles and responsibilities of all parties and best practices for communication. The need for standardized guidelines has been considered a priority in order to establish clinical utility [30,37,49,50,51].

Guidelines should provide instructions, implemented uniformly across clinics, on how to communicate PRS accurately avoiding potential harms [27,30,49,50,51]. A need for consistency and additional resources are imperative to ensure PRS comparability and evaluation, and therefore improve generalizability [52]. Guidelines will help to address ongoing challenges in implementation including, but not limited to, communication, population identification, and rights protection. 

#### 3.5.1. Communication Guidelines

Guideline development for PRS implementation in clinical practice is essential to ensure effective communication and minimize risk associated with communicating complex genetic information. Failures in risk communication can lead to misunderstandings, unwarranted health decisions, misplaced reassurance, and psychosocial patient impacts including anxiety or distress [37,53]. Guidelines can equip physicians with the tools to explain the intricate relationship between genetics, lifestyle, and environmental factors in disease etiology, helping patients to make informed decisions. Effective communication also necessitates a tailored approach, considering patients’ genomic literacy and educational background, and furthermore should be risk stratified. Personalized strategies that emphasize participatory communication and support patients’ personal utility in testing have shown to enhance comprehension and engagement [36,51]. Integrating PRS reporting with educational resources, like PRS score reports, online tools, or telemedicine consultations, can enhance understanding and promote positive health changes [54].

As PRS continues to grow and develop as a field, individuals may have their risk category shifted over time. Consideration must be taken to establish under what circumstances patients should be informed of these changes, and how to best communicate this possibility to patients [28,50]. How unknown and known information is communicated in a dynamic risk assessment tool should be integrated into guidelines to ensure an adaptable approach [28].

In resource-constrained or non-specialist settings, where the integration of PRS is often challenging, guidelines could streamline workflows by offering adaptable frameworks for disclosing results, including multidisciplinary approaches, as well as additional training and support for physicians [30,55,56,57]. A multidisciplinary approach can both help to alleviate resource constraints, and best utilize individuals’ skill sets. Genetic counselors play a pivotal role, not only in explaining the nuances of polygenic risk but also in addressing broader issues of social justice and public health that impact patient access to genomic services and appropriate risk-reducing interventions [51,55]. Other options for improving training and communication involve integrating genetic risk training into medical education, continuing education programs for physicians, and hiring genetic risk-assessment specialists who train physicians in interpreting and implementing data [50]. 

#### 3.5.2. Population Selection Guidelines

Guidelines are important to identify and apply PRS to the appropriate population to optimize their utility and ensure they are used effectively and equitably. One of the key considerations in applying PRS, particularly for CAD, is the age and baseline clinical risk of the individual. Younger patients, who have typically had fewer environmental exposures and lifestyle-related risk factors, are more likely to see PRS as a meaningful contributor to their long-term risk profile [58,59]. By identifying individuals with elevated genetic risk early in life, clinicians can facilitate proactive risk assessment and initiate preventive strategies before the onset of symptoms, maximizing the long-term benefits of PRS [50]. Conversely, in older adults with an established burden of risk factors, the relative influence of genetics diminishes, suggesting that routine PRS testing may have limited added value in this demographic [58]. This targeted approach aligns with recent findings that PRS for CAD should be applied selectively, with younger or intermediate clinical risk individuals gaining the most benefit, while indiscriminate testing may yield limited actionable insights [5,58,60].

Guidelines are also essential for ensuring the “responsible use” of PRS, as emphasized by the PRS Task Force, which advocates for applications where benefits clearly outweigh risks and equitable access is prioritized [37]. Clinical utility is a cornerstone in the debate over PRS integration in care, as it is critical to assess whether the use of PRS will genuinely enhance patients’ quality of life and contribute to improved health outcomes [50]. Developing guidelines to standardize these practices will help ensure that PRS testing is applied where it is most beneficial, avoiding overuse while maximizing patient-centered outcomes.

#### 3.5.3. Discrimination and Rights Protection

The need for clear guidelines addressing the ethical considerations and potential risks associated with PRS is increasingly pressing. Genetic information, while valuable for individual health insights, can also be accessed by entities such as employers and insurance companies, creating the risk of discrimination based on genetic predisposition [30]. The potential for misuse of genetic data underscores the importance of legislation, to protect individuals against discrimination in health insurance and employment. However, current legislation does not extend to life or disability insurance, leaving individuals vulnerable, highlighting the need for updated legislation that more comprehensively protects against genetic discrimination [15,19].

Guidelines also play a crucial role in addressing ethical questions surrounding the collection, sharing, and storage of genetic data. The increasing integration of artificial intelligence into PRS models, which utilize vast amounts of genomic and health data from electronic health records, raises new privacy and data protection concerns. As PRS models grow more complex, it becomes essential to reassess how these data are stored and protected to prevent misuse and ensure that individuals retain control over their genetic information [49,61]. Ethical considerations must also account for the right to access genomic data, the purposes for which it is used, and how it is safeguarded, especially given the potential for stigma or discrimination based on genetic predispositions [49].

### 3.6. Future Directions

PRS will continue to shape our clinical understanding of disease. Looking ahead, initiatives like the INNOPREV trial aim to advance personalized approaches to cardiovascular disease prevention, leveraging genetic insights to reduce disease burden and enhance public health outcomes [31]. While we continue to assess PRS and its many facets in utilization and implementation, there are a number of considerations that clinicians and researchers will have to grapple with. 

Further, the PRS Taskforce of International Common Disease has established both short- and long-term objectives to inform responsible use of PRS by establishing benefits (e.g., determining clinical utility and adopting professional standards), mitigating risks (e.g., improving representation of diverse populations), and closing gaps (e.g., enabling translational applicability) [37]. These objectives can be used broadly to guide future efforts in development of PRS implementation and use. 

There is a pressing need to develop guidelines prior to the integration of PRS into clinical practice broadly, and beyond subspecialty clinics. This is essential to provide physicians with the training and tools they need to address patient and population concerns and knowledge in how to best utilize health resources. Table 1 summarizes the challenges currently faced in PRS implementation into clinical practice. While PRS is a promising and clear future direction of genetic stratification, care must first be taken to establish how best to integrate it into our healthcare system. 

## 4. Conclusions

The integration of PRS into clinical practice offers an innovative approach to CVD risk assessment, enhancing predictive accuracy and enabling earlier, more personalized interventions. While traditional models have proven instrumental for CVD risk assessment, they are limited by their reliance on non-genetic factors. PRS offers a complimentary resource, providing insight into genetic predispositions that aid in the generation of robust risk assessments. Table 2 highlights the many facets of PRS implementation discussed in this review as well as some considerations for the future.

As research advances, PRS demonstrates promise in refining CVD risk predictions, especially when combined with established clinical models. Despite this potential, significant challenges remain, particularly regarding the practical and ethical implications of PRS implementation. Effective guidelines are essential to ensure standardized practices, support patient and provider education, and address the ethical concerns of privacy and discrimination. 

Integration into clinical practice also depends on addressing both patient and provider experience. While PRS can be seen as a tool to support patient’s health literacy and self-efficacy, care must be taken to address patient anxiety, emphasizing the need for supportive communication. Providers support PRS but require standardized guidelines and training to implement it effectively. Developing these resources will be essential for PRS to fulfill its potential in personalized, patient-centered care. By addressing these barriers, PRS may ultimately serve as a transformative tool in preventive cardiology, fostering a proactive approach to disease management that could improve patient outcomes and advance personalized medicine.

## Figures and Tables

**Figure 1 genes-15-01581-f001:**
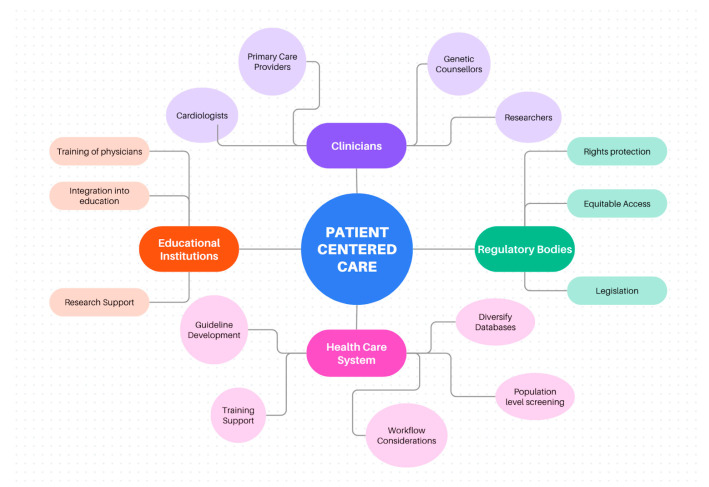
Representation of each of the key stakeholders and their involvement in and influence on PRS implementation into clinical practice.

**Table 1 genes-15-01581-t001:** Summary of challenges faced by PRS integration into clinical practice and proposed solutions.

Challenge	Description	Proposed Solutions
Lack of Diversity in Genomic Data	Most PRS are derived from studies in populations of European ancestry, limiting applicability to other groups [5,8,30]	Conduct multi-ethnic GWAS, adjust PRS for diverse populations, and include ancestry-specific datasets [7,12]
Communication of Results	Complexity in explaining PRS and associated risks to patients, leading to potential misinterpretation or anxiety [37,53]	Provide tailored, clear explanations with visual aids; involve genetic counselors to address concerns effectively [28,36,51,54]
Ethical and Privacy Concerns	Potential misuse of genetic information by insurers, employers, or other entities [19,30]	Strengthen legal protections (e.g., expand GINA); develop secure data storage and sharing protocols [8,15,19]
Limited Provider Training	Many providers lack knowledge and confidence in interpreting and using PRS results in clinical care [44,47,48]	Integrate genetic training into medical education; offer continuing education and decision-support tools [50,51,55]
Dynamic Nature of PRS	As genetic data evolves, risk scores may change, leading to patient confusion and need for re-assessment [28,50]	Establish guidelines for re-assessment; communicate the dynamic nature of genetic risk to patients [28]
Limited Actionability	Lack of clear guidelines on actionable steps based on PRS results [48]	Develop standardized clinical guidelines; define pathways for using PRS in preventive and therapeutic decisions [3,5,60]

**Table 2 genes-15-01581-t002:** Summary of key aspects of PRS addressed in this paper, the associated challenges, and future directions.

Aspect	Key Insights	Challenges	Future Directions
Clinical Applications	Improves risk prediction for CAD, T2D, AF, and prostate cancer Enhances conventional risk models	Limited validation across diverse populations Modest improvements in some contexts	Conduct multi-ethnic studies Incorporate environmental and epigenetic factors
Integration into Practice	Useful for early diagnosis, treatment stratification, and lifestyle interventions	Lack of guidelines Workflow integration barriers	Develop standardized communication and implementation guidelines Incorporate genetic counseling
Patient Perspectives	Improves perceived control over healthMotivates lifestyle changes Generally understandable	Numerical complexity Potential for anxiety or negative emotions	Enhance educational tools Tailor communication to literacy levels Address emotional concerns
Provider Perspectives	Feasible and clinically useful Supports preventive behavior Identifies high-risk individuals	Need for training Ambiguity in actionable steps Workflow adaptation	Incorporate PRS into medical education Provide clear guidelines Multidisciplinary collaboration
Ethical Considerations	Enhances personalized medicine Informs proactive health management	Risk of genetic discrimination Privacy Concerns	Strengthen genetic data protection laws Address equity in accessEnsure informed consent

## Data Availability

Data sharing is not applicable.

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
