# Peer review of "Polygenic Risk Score Implementation into Clinical Practice for Primary Prevention of Cardiometabolic Disease"

_genes, 2024, doi:10.3390/genes15121581_

Round 1

Reviewer 1 Report

Comments and Suggestions for Authors

The authors of this review paper provided an overview of PRS in Cardiometabolic traits, and how it can be utilized in preventive care. Overall, the paper is well written and the previous literature clearly discussed. These are my comments:

  1. The first sentences in the abstract are too specific for Canada. Since this specific population is not the focus of this review, I believe more general claims are needed.

  2. Sentence in lines 108-109 needs a reference.

  3. Since the focus of the review is Cardiometabolic disorders, the paragraph starting at line 145 (about prostate cancer patients) could be better connected, or replaced by a Cardiometabolic disorder.

  4. There are some terms that might need more explanation: Gleason scores (line 149), Concordance Index (line 153).

  5.  INNOPREV study needs a reference if available

  6. In section 3.2 it would be interesting to have a small summary of the contribution of rare variants in CVD risk. Or even the integration of common and rare variants in the prs construction.

Author Response

Comments 1: The first sentences in the abstract are too specific for Canada. Since this specific population is not the focus of this review, I believe more general claims are needed.

Response 1: Thank you for pointing this out, we agree with this comment and therefore have included more globally applicable statement found in the first sentences of the abstract. This change can be found in page 1, paragraph 1, line 11 and 15 (World Heart Report, 2023)

Comment 2: Sentence in lines 108-109 needs a reference.

Response 2:  Thank you for pointing this out, we agree and have added references in support of this statement in lines 108 to 109. This change can be found on page 3 in the 2nd paragraph on line 116. 

Comments 3: Since the focus of the review is Cardiometabolic disorders, the paragraph starting at line 145 (about prostate cancer patients) could be better connected, or replaced by a Cardiometabolic disorder.

Response 3: Thank you for pointing this out,we agree with this comment. Therefore, we have removed this example. This change occurred on page 4 line 152.

Comment 4: There are some terms that might need more explanation: Gleason scores (line 149), Concordance Index (line 153).

Response 4: Thank you for pointing this out, we agree with this comment. Therefore, we have added a brief explanation for the Concordance index on page 4, line 154-155. We have removed the example containing Gleason scores based on relevance and therefore this term is no longer used in the manuscript.

Comment 5: INNOPREV study needs a reference if available

Response 5: Thank you for pointing this out, we agree with this comment and have added the relevant reference for the INNOPREV study. This change can be found on page 4, line 165.

Comment 6: In section 3.2 it would be interesting to have a small summary of the contribution of rare variants in CVD risk. Or even the integration of common and rare variants in the prs construction.

Response 6: Thank you for pointing this out, we agree with this comment. Therefore, we have added additional findings from a relevant study into that section. These findings nicely incorporate the contribution of rare variants in PRS construction. This change can be found on page 5, lines 195-199.

Reviewer 2 Report

Comments and Suggestions for Authors

Multi Digital Publishing Institute (MDPI): genes- 3349057

Title: Review

Polygenic Risk Score Implementation into Clinical Practice for Primary Prevention of Cardiometabolic Disease

Overall comments:

This may be an important and intriguing topic in the era of precision medicine and genomic medicine.  

This is a review paper on clinical implementation of polygenic risk score of predicting (primary prevention) cardiometabolic disease.  Overall, it is well written and covers many aspects of the use of PRS.  

However, it may be important to determine whether the authors are aiming to discuss PRS for cardiovascular disease (CVD) or cardiometabolic disease.  Although these disease groups are often used and interpreted interchangeably, they should be considered slightly different from the point of view of clinical genetics.

Cardiovascular disease – vascular system centric perspective which Includes heart attack, stroke, angina (chest pain), and other disorders of the vascular system. 

Cardiometabolic disease – more metabolism and biochemical including other factors which can be described as cardiovascular diseases, plus other conditions such as type 2 diabetes, insulin resistance, and non-alcoholic fatty liver disease. 

Cardiometabolic diseases are a leading cause of death worldwide and are increasing rapidly. Risk factors for cardiometabolic diseases include:

Obesity

Smoking

Hypertension

Insulin resistance

High fasting triglycerides

Low "good" HDL cholesterol 

The combination of abdominal obesity, high fasting triglycerides, low "good" HDL cholesterol, and elevated blood pressure is often referred to as metabolic syndrome. 

Although the distinction may be very subtle, and not many people consider the importance of separating them, it is important that the authors consider when reviewing such topic.

Line 65: please review the definition of “penetrance” in clinical genetics.

https://medlineplus.gov/genetics/understanding/inheritance/penetranceexpressivity/#:~:text=Penetrance%20refers%20to%20the%20proportion,genetic%20condition%20to%20future%20generations.

“Incomplete penetrance” and “variable expressivity” are two confusing words, and “penetrance” is often a misused word.

From the authors’ statement, the authors means that some “expression” of a monogenic variant may be reduced?

Incomplete penetrance means that the authors are stating that a monogenic variant is not expressed at all.

This word is often misused even in a reputable journal, and we have to start correcting this whenever possible.

It is possible that monogenic variants are expressed (penetrant), but due to other influences such as the polygenic background so that it may seem as though it is non-penetrant (not expressed). 

Please consider some concerns with the development of a PRS scheme below:

1) CVD or cardiometabolic disease itself is tremendously heterogeneous.  It probably has different etiologies in each individual.  It may be important that the definition or category of CVD or cardiometabolic disease is to be refined before a PRS can be applied, in addition to racial background categorization.

ï‚§ Cardiovascular disease (important to include vascular components in addition)

ï‚§ Cardiometabolic disease (important to include metabolic derangements in addition)

2) Commonly used risk factors such as diabetes, total cholesterol, HDL-cholesterol, and blood pressure have its underlying genetic underpinning.

3) The need to identify monogenic variants which may have a stronger influence on the resulting phenotype prior to the application of a PRS?  Go through the actionable variant identification first?

4) As the authors pointed out.

The importance of genetic education since many physicians have not received enough through medical school etc. so that some may not really understand the meaning of SNPs or CNVs.  It is important that many SNPs used in GWAS may or may not have any biological role, and it is uncertain that even with a high statistical significance, it has an important biological role.  A high statistical significance can never replace a valid result of functional studies.

5) It may be possible that some SNPs are susceptibility SNPs rather than inherent risk SNPs.

Agree with ethical and privacy issues and education in Table 1.

It is important that the addition of PRN outperforms the use of traditional risk factors.

FYI:

Actionable secondary findings (via WES or WGS)

Miller DT, Lee K, Abul-Husn NS, Amendola LM, Brothers K, Chung WK, Gollob MH, Gordon AS, Harrison SM, Hershberger RE, Klein TE, Richards CS, Stewart DR, Martin CL; ACMG Secondary Findings Working Group. Electronic address: documents@acmg.net. ACMG SF v3.2 list for reporting of secondary findings in clinical exome and genome sequencing: A policy statement of the American College of Medical Genetics and Genomics (ACMG). Genet Med. 2023 Aug;25(8):100866. doi: 10.1016/j.gim.2023.100866. Epub 2023 Jun 22. PMID: 37347242; PMCID: PMC10524344.

Thank you very much for allowing me to review this manuscript.  

Sincerely, 

Author Response

Comment 1: 

This is a review paper on clinical implementation of polygenic risk score of predicting (primary prevention) cardiometabolic disease.  Overall, it is well written and covers many aspects of the use of PRS.  

However, it may be important to determine whether the authors are aiming to discuss PRS for cardiovascular disease (CVD) or cardiometabolic disease.  Although these disease groups are often used and interpreted interchangeably, they should be considered slightly different from the point of view of clinical genetics.

• Cardiovascular disease – vascular system centric perspective which Includes heart attack, stroke, angina (chest pain), and other disorders of the vascular system. 

• Cardiometabolic disease – more metabolism and biochemical including other factors which can be described as cardiovascular diseases, plus other conditions such as type 2 diabetes, insulin resistance, and non-alcoholic fatty liver disease. 

Cardiometabolic diseases are a leading cause of death worldwide and are increasing rapidly. Risk factors for cardiometabolic diseases include:

• Obesity

• Smoking

• Hypertension

• Insulin resistance

• High fasting triglycerides

• Low "good" HDL cholesterol 

The combination of abdominal obesity, high fasting triglycerides, low "good" HDL cholesterol, and elevated blood pressure is often referred to as metabolic syndrome. 

Although the distinction may be very subtle, and not many people consider the importance of separating them, it is important that the authors consider when reviewing such topic.

Response 1: Thank you for identifying this gap in clarity of language. We have addressed this concept in section 1.3, (page 3, lines 112-114) and addressed this throughout the paper to ensure accuracy of our statements.

Comment 2: 

Line 65: please review the definition of “penetrance” in clinical genetics.

https://medlineplus.gov/genetics/understanding/inheritance/penetranceexpressivity/#:~:text=Penetrance%20refers%20to%20the%20proportion,genetic%20condition%20to%20future%20generations.

“Incomplete penetrance” and “variable expressivity” are two confusing words, and “penetrance” is often a misused word.

From the authors’ statement, the authors means that some “expression” of a monogenic variant may be reduced?

Incomplete penetrance means that the authors are stating that a monogenic variant is not expressed at all.

This word is often misused even in a reputable journal, and we have to start correcting this whenever possible.

It is possible that monogenic variants are expressed (penetrant), but due to other influences such as the polygenic background so that it may seem as though it is non-penetrant (not expressed). 

Response 2: Thank you for pointing this out. We agree with your comment and have determined that the meaning of this statement is more accurately described with the term “variable expressivity”, as its expression is related to a polygenic background. This phrasing has been changed to reflect this variable expressivity in line 75 page 2.

Comment 3: 

Please consider some concerns with the development of a PRS scheme below:

1) CVD or cardiometabolic disease itself is tremendously heterogeneous.  It probably has different etiologies in each individual.  It may be important that the definition or category of CVD or cardiometabolic disease is to be refined before a PRS can be applied, in addition to racial background categorization.

ï‚§ Cardiovascular disease (important to include vascular components in addition)

ï‚§ Cardiometabolic disease (important to include metabolic derangements in addition)

2) Commonly used risk factors such as diabetes, total cholesterol, HDL-cholesterol, and blood pressure have its underlying genetic underpinning.

3) The need to identify monogenic variants which may have a stronger influence on the resulting phenotype prior to the application of a PRS?  Go through the actionable variant identification first?

4) As the authors pointed out.

The importance of genetic education since many physicians have not received enough through medical school etc. so that some may not really understand the meaning of SNPs or CNVs.  It is important that many SNPs used in GWAS may or may not have any biological role, and it is uncertain that even with a high statistical significance, it has an important biological role.  A high statistical significance can never replace a valid result of functional studies.

5) It may be possible that some SNPs are susceptibility SNPs rather than inherent risk SNPs.

Response 3: 

Thank you for pointing this out. We agree with your comments, therefore we have attempted to incorporate considerations of these factors in the following capacities. 

We have further clarified our usage of the terms CVD vs Cardiometabolic disease (page 3, lines 112-114 ) and edited for consistency throughout the paper to ensure accuracy of our statements

We have noted the underlying genetic underpinning of some commonly used risk factors on page 2 lines 57-58.

We have added the consideration to our PRS definition that many SNPs may or may not have a biological role on page 2 lines 71-74.

We have added mention of the need to identify influential monogenic variants on page 5 lines 210-211.